# Rapamycin Alternatively Modifies Mitochondrial Dynamics in Dendritic Cells to Reduce Kidney Ischemic Reperfusion Injury

**DOI:** 10.3390/ijms22105386

**Published:** 2021-05-20

**Authors:** Maria Namwanje, Bijay Bisunke, Thomas V. Rousselle, Gene G. Lamanilao, Venkatadri S. Sunder, Elizabeth C. Patterson, Canan Kuscu, Cem Kuscu, Daniel Maluf, Manjari Kiran, Valeria Mas, James D. Eason, Amandeep Bajwa

**Affiliations:** 1Department of Pediatrics, The University of Tennessee Health Science Center, Memphis, TN 38103, USA; mnamwanj@uthsc.edu; 2Transplant Research Institute, James D. Eason Transplant Institute, Department of Surgery, College of Medicine, The University of Tennessee Health Science Center, Memphis, TN 38163, USA; glamani1@uthsc.edu (G.G.L.); epatte18@uthsc.edu (E.C.P.); ckuscu@uthsc.edu (C.K.); ckuscu1@uthsc.edu (C.K.); jeason1@uthsc.edu (J.D.E.); 3Department of Genetics, Genomics, and Informatics, College of Medicine, The University of Tennessee Health Science Center, Memphis, TN 38163, USA; bbisunke@uthsc.edu; 4Department of Surgery, Surgical Sciences Division, School of Medicine, University of Maryland, Baltimore, MD 21201, USA; trousselle@som.umaryland.edu (T.V.R.); vmas@som.umaryland.edu (V.M.); 5Department of Systems and Computational Biology, School of Life Sciences, University of Hyderabad, Hyderabad 500046, India; shyamuohyd@gmail.com (V.S.S.); manjari.hcu@uohyd.ac.in (M.K.); 6Division of Transplant Surgery, University of Maryland Medical Center, School of Medicine, University of Maryland, Baltimore, MD 21201, USA; dmaluf@som.umaryland.edu; 7Department of Microbiology, Immunology, and Biochemistry; College of Medicine, The University of Tennessee Health Science Center, Memphis, TN 38163, USA

**Keywords:** dendritic cell, rapamycin, mitochondria, acute kidney injury, ischemic reperfusion injury

## Abstract

Dendritic cells (DCs) are unique immune cells that can link innate and adaptive immune responses and Immunometabolism greatly impacts their phenotype. Rapamycin is a macrolide compound that has immunosuppressant functions and is used to prevent graft loss in kidney transplantation. The current study evaluated the therapeutic potential of *ex-vivo* rapamycin treated DCs to protect kidneys in a mouse model of acute kidney injury (AKI). For the rapamycin single (S) treatment (Rapa-S-DC), Veh-DCs were treated with rapamycin (10 ng/mL) for 1 h before LPS. In contrast, rapamycin multiple (M) treatment (Rapa-M-DC) were exposed to 3 treatments over 7 days. Only multiple *ex-vivo* rapamycin treatments of DCs induced a persistent reprogramming of mitochondrial metabolism. These DCs had 18-fold more mitochondria, had almost 4-fold higher oxygen consumption rates, and produced more ATP compared to Veh-DCs (Veh treated control DCs). Pathway analysis showed IL10 signaling as a major contributing pathway to the altered immunophenotype after Rapamycin treatment compared to vehicle with significantly lower cytokines *Tnfa, Il1b, and Il6,* while regulators of mitochondrial content *Pgc1a*, *Tfam*, and *Ho1* remained elevated. Critically, adoptive transfer of rapamycin-treated DCs to WT recipients 24 h before bilateral kidney ischemia significantly protected the kidneys from injury with a significant 3-fold improvement in kidney function. Last, the infusion of DCs containing higher mitochondria numbers (treated *ex-vivo* with healthy isolated mitochondria (10 µg/mL) one day before) also partially protected the kidneys from IRI. These studies demonstrate that pre-emptive infusion of *ex-vivo* reprogrammed DCs that have higher mitochondria content has therapeutic capacity to induce an anti-inflammatory regulatory phenotype to protect kidneys from injury.

## 1. Introduction

The pathobiology of acute kidney injury is multifactorial and involves intricate interactions between renal parenchymal cells and immune cells [1,2]. In the kidney, conventional dendritic cells (DCs) are the major immune cell subset and are found intimately in the interstitial space throughout the kidney [3,4,5]. DCs are one of the primary cell types involved in innate immunity and are essential in providing protection against pathogens. Our previously published work using bone marrow derived DCs has demonstrated that DCs therapy can regulate various innate and adaptive immune responses associated with AKI [6,7].

DCs are located extensively throughout all tissues, with a subset of DCs (CD11c^+^ DCs) residing in the renal parenchyma. However, these CD11c^+^ DCs also play a pathogenic role in worsening renal injury following ischemia-reperfusion injury (IRI) [8]. IRI is defined as injury resulting from the reintroduction of oxygenated blood to an organ after a period of ischemia and is often seen in myocardial infarction, stroke, and solid organ transplantation. In response to IRI, CD11c^+^ DCs comprise a considerable portion of the inflammatory cell infiltrate in renal tissue.

DCs are capable of recognizing pathogen-associated molecular patterns (PAMPs) through various pattern recognition receptors located on the surface of DCs such as toll-like receptors (TLRs) and NOD-like receptors. Binding of pattern recognition receptors (PRRs) to their specific PAMPs results in the activation of several pathways, which include proinflammatory responses, antimicrobial peptide synthesis, and induction of lymphocyte-mediated adaptive immunity [9]. Significant research has been dedicated to studying changes in the cellular bioenergetics of DCs upon activation by PAMPs. It has been established that activation of DCs with lipopolysaccharide (LPS; a PAMP distinct to gram-negative bacteria), via binding to its specific receptor TLR4 results in a metabolic change from highly efficient oxidative phosphorylation to a process very similar to Warburg metabolism, or aerobic glycolysis. Warburg metabolism is characterized by diverting pyruvate generated from glycolysis towards conversion into lactate for energy production, and away from transport into the mitochondria to undergo catabolism via the TCA cycle, despite sufficient oxygen availability. One explanation for this phenomenon involves one of the many downstream effects of TLR4 activation, which is the increased activity of the serine/threonine kinase the mammalian target of rapamycin (mTOR). mTOR inhibition by rapamycin preserves mitochondrial function in activated DCs by downregulating iNOS, with rapamycin-treated DCs stimulated in the presence of LPS exhibiting reduced levels of aerobic glycolysis [10]. mTORC1 inhibitors like rapamycin are currently used in solid organ transplantation as a possible alternative for calcineurin inhibitors to avoid chronic renal allograft damage as it also has potential to expand of CD4^+^CD25^+^ regulatory T-cells [11,12]. However, rapamycin use in transplantation can lead to inflammatory complications that include peripheral edema, creatinine increases, and thrombocytopenia. Previous studies have demonstrated that the adverse effects of hypoxic stimuli are enhanced by LPS exposure and attenuated by rapamycin [8,13].

The goal is to develop methods to use FDA approved clinical grade drugs like Rapamycin to induce regulatory DCs that can control unwanted innate and adaptive immune responses. The aim of this study was to determine the potential protective mechanism(s) of rapamycin stimulated BMDCs in a preclinical mouse model of kidney IRI. Treatment of BMDCs *ex-vivo* with rapamycin avoids any adverse off-target effects and inflammatory complications associated with systemic drug injections. The optimal dose and timing of most pharmacological compounds or drugs are dependent on various parameters. In cancer cells low-dose of rapamycin compared to high-dose differentially inhibits the mTORC1 and mTORC2 resulting in either partial or complete inhibition of cell-cycle progression [14]. For all of ours *ex-vivo* BMDCs conditioning studies in current manuscript we chose the low dose of 10 ng/mL of rapamycin based on previously reported studies with tolerogenic BMDCs [15,16]. The timing of how many doses (three or one) was based on our previously published work with propagating mouse BMDCs and added with GMCSF [17,18]. Our current data demonstrates that both single and multiple treatment(s) of DCs with rapamycin induces less robust immunogenic responses (lower cytokines and co-stimulatory molecules) after LPS stimulation. Transfer of either single or multiple rapamycin treated DCs equally protects kidneys from IRI in syngeneic (C57BL/6 BMDC→C57BL/6 mice) or allogeneic (BALB/c BMDC→C57BL/6 mice) models of IRI. However, multiple *ex-vivo* treatment with rapamycin (3 treatments over 7 days), in addition to reducing the immunogenic responses, also increased mitochondria numbers and function (Seahorse Analyzer) compared to vehicle-treated DCs. These experimental observations suggest DCs with higher mitochondria numbers have the potential to become regulatory, and the transfer of theses DCs may protect kidneys from ischemic injury. As our recent study using FTY720 treated DCs demonstrated [18], pharmacological or mitochondrial transplants that induce higher mitochondria numbers in immune cells (DCs or macrophages) also cause these cells to have an anti-inflammatory phenotype, an effect possibly due to maintenance of mitochondria oxidative phosphorylation (OXPHOS) and with reduced switch to glycolysis for energy after stimulation. To these effects, we exogenously increased mitochondria numbers in DCs to test if higher mitochondria numbers can induce a regulatory phenotype by using mitochondrial transplants. Transfer of these Mito-DCs protected kidneys from IRI; however, unlike Rapa-DCs these Mito-DCs do not have reduced immunogenic responses (MHCII, co-stimulatory molecules and cytokines) after LPS stimulation, possibly indicating that increasing mitochondria copy number can override functional changes induced by exogenous LPS stimulation. In this study and as previously demonstrated [6,7,18], the systemically injected DCs (vehicle-, rapamycin- or mitochondria-treated) at the dose of half-million are predominantly only found in spleen and little to no signal is found in the kidneys. In spleen like FTY720-DCs, rapamycin and mitochondria rich DCs may induce an anti-inflammatory response resulting in protection from kidney IRI. These seminal findings provide novel evidence that suggests increasing mitochondrial numbers (exogenously through mitochondrial transplants or via pharmacological intervention (FTY720 or rapamycin)) in immune cells can be a new therapeutic modality to ultimately regulate their immunologic responses for prevention and/or early treatments of kidney injury. 

## 2. Materials and Methods

### 2.1. Mice

All animal procedures and handling were performed in adherence to the National Institutes of Health Guide for the Care and Use of Laboratory Animals, and Bajwa UTHSC protocol number, 18-080, approval date 09/28/2018 was approved by the University of Tennessee Health Science Center (UTHSC) Institutional Animal Care and Use Committees. Laboratory animal care policies follow the Public Health Service Policy on Humane Care and Use of Laboratory Animals. All results are reported in a manner consistent with ARRIVE guidelines [19]. For all transfer studies C57BL/6J and BALB/c mice were purchased from The Jackson Laboratory (Bar Harbor, ME, USA). Mice were maintained in standard vivarium housing with a 12-hr light/dark cycle on a chow diet and water was freely available.

### 2.2. Renal Ischemia-Reperfusion Injury

All detailed surgical procedures were followed as previously reported [18]. C57BL/6 male mice (8–12 weeks old) were subjected to 26-min bilateral ischemia followed by 20–24 h reperfusion. Briefly, all male mice were anesthetized with an intraperitoneal (i.p.) injection of a ketamine and xylazine mixture, with subcutaneous injection of buprenorphine and placed on a warm pad to maintain body temperature at 34.5–36 °C. Mice were then randomized to the sham or IRI operation condition. Bilateral flank incision was performed. Body temperature was checked and maintained throughout the ischemic period. Sham-operated mice underwent the same procedure except for vessel clamping and closing of surgical wounds. Mice that had one kidney with no reperfusion 24 h after ischemia were excluded from all analysis.

### 2.3. Assessment of Kidney Function and Histology

Blood was collected under anesthesia from the retro-orbital sinus, and plasma creatinine (mg/dL) was determined by using an enzymatic method with minor modifications from the manufacturer’s protocol (Diazyme Laboratories, Poway, CA, USA) and blood urine nitrogen (BUN) and creatinine measurement was performed using a QuantiChrom Urea Assay Kit (DIUR-100, BioAssay Systems, Hayward, CA, USA) as previously described [18,20]. For histology, kidneys were fixed over-night in 4% PLP or 10% formalin and embedded in paraffin. Kidneys were prepared for H&E staining as previously described [18] and photographs were taken with brightness/contrast adjustment made with the EVOS microscope (Thermo Fisher Scientific, Waltham, MA, USA). For quantification of tubular injury score, sections were assessed by counting the percentage of tubules that displayed cell necrosis, loss of brush border, cast formation, and tubule dilation as follows: 0 = normal; 1 = ≤10%; 2 = 10 to 25%; 3 = 26 to 50%; 4 = 51 to 75%; 5 = ≥75%. Five to 10 fields from each outer medulla were evaluated and scored in a blinded manner. The histological change was expressed as acute tubular necrosis (ATN), scored as previously described [7,21]. Apoptotic cells were detected by TUNEL assay (In Situ Cell Death Detection kit, TMR Red; Roche, Basel, Switzerland) according to the manufacturer’s instructions.

### 2.4. Bone Marrow (BM)-Derived-Dendritic Cell (DC) Culture and Adoptive Transfer

8–10 weeks old C57BL/6J or BALB/c WT male mice were used for generating DCs from whole BM precursors [22]. GMCSF-rich supernatant was derived from J558L cells stably transfected with mouse GMCSF. The cell line was a generous gift from Dr. Ira Mellman (Department of Biology, Yale University, New Haven, CT, USA). Briefly, freshly isolated BM was cultured with 6 ng/mL recombinant mouse GMCSF (total of 3 treatments) for 8 days in RPMI 1640 (Invitrogen, Carlsbad, CA, USA). The optimal dose of 10 ng/mL rapamycin was decided after testing various doses (0.1–10 ng/mL). BMDC were treated with 10 ng/mL rapamycin (Sigma, St. Louis, MO, USA) for a total of three treatments or 1-overnight treatment). BMDCs were treated with the TLR4 agonist lipopolysaccharide (LPS; *Escherichia coli* serotype 0111:B4: 100 ng/mL; Sigma-Aldrich); or vehicle (1 × PBS) for 24 h in culture medium for syngeneic studies (C57BL/6J BMDCs→C57BL/6J mice) and left untreated for allogenic studies (BALB/c BMDCs→C57BL/6J mice). Cells were washed, and 0.5 × 10^6^ cells per mouse were intravenously (i.v.) injected into naive mice 1 day before bilateral kidney IRI. BMDCs were labeled with MitoTracker CMXRos Red or MitoTracker Deep Red (100 nM, 30 min @ 37 °C, Invitrogen) or Mitosox (5 µM; 10 min @ 37 °C; Invitrogen).

### 2.5. Quantitative Real-Time PCR

Relative mitochondrial DNA (mtDNA) expression level was measured as previously described [23]. Briefly, total genomic DNA was isolated and equal amounts (5 ng) was used for RTPCR using ND1 as surrogate primers for mtDNA and HK2 primers for nuclear DNA (nDNA) for mouse and ND6 for human mtDNA as previously described [23,24]. Total RNA was isolated and reversed transcribed to cDNA, and RT-PCR was performed as previously described [7,25,26].

### 2.6. Mitochondria Isolation and Quantification

Mitochondria were isolated from HEK293 cells as previously described [18]. Briefly, 60 cm plates of HEK293 cells were collected by adding 5 mL of homogenization buffer (300 mmol/L sucrose, 10 mmol/L HEPES-KOH, 1 mmol/L EGTA-KOH, pH 7.4) with 1 mg Subtilisin A protease from *Bacillus licheniformis* (Sigma-Aldrich). A detailed protocol for mitochondria isolation can be found in our previous published manuscript [18].

### 2.7. Seahorse Flux Bioanalyzer

Seven-day old BMDCs were transferred to a Seahorse 24-well tissue culture plates, and oxygen consumption rate (OCR) was measured, and parameters were calculated as previously described [25] with the following modification. After measuring basal respiratory rate, oligomycin (Sigma; 2 μM), FCCP (Sigma; 1.5 μM; carbonyl cyanide 4-(trifluoromethoxy)-phenylhydrazone (FCCP)), and electron transport chain (complex I and III) inhibitors, rotenone (Sigma; 0.5 μM), and antimycin A (Sigma; 0.5 μM)) were injected sequentially during the assay. Basal mitochondrial respiration, ATP-linked respiration, proton leak (non-ATP linked oxygen consumption), maximal respiration, non-mitochondrial respiration, reserve respiratory capacity, respiratory control ratio, and coupling efficiency were determined in whole cells according to Brand et al. [27]; N = 4–5 wells were used for each experimental group and experiments were repeated a minimum of 3 times.

### 2.8. Flow Cytometric Analysis, Western Blot, and ELISA

Flow cytometry data acquisition was performed on a FACS Calibur (Becton Dickinson, San Jose, CA, USA) with Cytek 8-color flow cytometry upgrade (Cytek Development, Inc., Fremont, CA, USA). Data were analyzed by FlowJo software 9.0 (Tree Star, Ashland, OR, USA) as previously described [6,7,28]. For ELISA media was collected from BMDCs 24 h after treatment with 100 ng/mL LPS. TNFα, IL6, or IL10 levels were measured by using mouse ELISA kits (Invitrogen) following the manufacturer’s protocol and as previously described [18]. Veh- or Mito-DC treated with and without LPS for 6 or 24 h were used to isolated total protein using RIPA lysis buffer supplemented with protease and phosphatase inhibitor cocktail (Thermo Fisher Scientific, Vernon Hills, IL, USA). For GAPDH lysate supernatants were either boiled (10 min at 100 °C) and lysates were left at room temperature for 10 min for rodent OXPHOS cocktail (Abcam, Cambridge, MA, USA) [18]. A more detailed methods have been previously described [18].

### 2.9. Data and Statistical Analysis

GraphPad Prism 8 (GraphPad Inc., San Diego, CA, USA) was used to analyze and present the data. Data were analyzed, after transformation if needed to generate a normal distribution, by 2-tailed *t* test or 1-way ANOVA with post-hoc analysis as appropriate. Two-tailed unpaired *t* test was used for analysis of two groups. *p*≤ 0.05 was used to indicate significance.

## 3. Results

### 3.1. Multiple Treatments with Rapamycin (Rapa-M) Induces Higher Mitochondria Numbers and Less Immunogenic DCs

C57BL/6 WT DCs were isolated and propagated for 8 days in presence of GMCSF and vehicle (1X PBS) or rapamycin (10 ng/mL), total of three treatments. DCs were treated with 100 ng/mL LPS over-night. LPS treated DCs were used for RNASeq or labeled with various mitochondrial dyes. The RNASeq data confirmed previous data that rapamycin treatment induces less immunogenic DCs with lower pro-inflammatory cytokines that regulate innate and adaptive immune responses with IL10 as the major changing pathway (Appendix A). Compared to vehicle treatment, rapamycin treatment increased mitochondrial content in BMDCs (Figure 1A) as checked with MitoTracker CMXRos Red (50 nM) with and without LPS. Similarly, there was significantly higher labeling with MitoTracker Deep Red (100 nM) and significantly lower labeling with MitoSox (5 µM) and after over-night LPS stimulation in Rapa-M-DC compared to Veh-DC (Figure 1B,C). Semi-quantification of histograms MFI for all mitochondrial dyes (Figure 1D). Additionally, Veh- and Rapa-M-DCs were staining using JC-1 to check if Rapamycin treatment changed the mitochondria membrane potential. Multiple treatment with rapamycin does not change the baseline mitochondria membrane potential compared to vehicle treated control BMDCs as shown by immunofluorescence (Appendix A) and flow cytometry (Appendix A).

Altered mitochondrial function, bioenergetic analysis was evaluated using Seahorse Bioanalyzer. As previously reported [18], LPS blunted oxygen consumption compared to Veh controls in Veh-DCs as expected (Figure 1E, blue to black line). Interestingly, rapa-M-DCs with or without LPS have higher basal OCR (Figure 1E, green to blue line at time zero). Rapa-M-DC demonstrated a failure to increase maximal respiratory capacity after FCCP injection in unstimulated cells that was maintained with LPS stimulation, demonstrating that rapamycin ablates spare respiratory capacity, as our previously reported work with FTY-DC [18]. LPS treated DCs had reduced ATP production as measured by OCR (blue to black) compared to Veh-DCs. Rapa-M-DC demonstrated significantly greater ATP production compared to Veh-DC in both unstimulated and LPS-stimulated DCs (Figure 1F). Rapa-M-DCs displayed elevated mRNA levels for peroxisome proliferator-activated receptor gamma co-activator 1-alpha (*Pgc1a*) compared to Veh-DCs (Figure 1G) and gene expression for *Pgc1a* and mitochondrial transcription factor A (*Tfam*) were maintained in Rapa-M-DCs after LPS stimulation. These data indicate that propagation of BMDCs in the presence of rapamycin increased mitochondrial content, basal OCR, and ATP production.

To test if rapamycin in addition to increasing mitochondria numbers and function also induced a change in DC co-stimulatory molecules after LPS was tested. Interestingly, after LPS treatment, Rapa-M-DC have significantly lower expression levels of co-stimulatory antigen presentation molecules (CD80, CD86 and CD40), MHCII, and PDL1 compared to Veh-DC (Figure 1H, blue vs. red). Interestingly, Rapa-M-DCs had similar expression levels for PDL1 compared to Veh-DCs but had a significant increase in PDL1/CD86 ratio after LPS stimulation compared to LPS treated Veh-DCs where the PDL1/CD86 decreased (Veh/Veh 0.59 ± 0.03 vs. Veh/LPS 0.43 ± 0.002, *p* < 0.001 and Rapa/Veh 0.56 ± 0.001 vs Rapa/LPS 0.73 ± 0.02, *p* < 0.05). No significant changes in CD11c expression were observed between Veh- and Rapa-M-DCs, although Rapa-M-DC had lower side scatter signal indicating the smaller size of cells after LPS stimulation (data not shown). Rapa-M-DCs had higher relative mtDNA/nDNA levels compared to Veh-DCs (105.7 ± 8.7 vs 196.8.9 ± 49.3), demonstrating rapamycin treatment increases DC mitochondria numbers. rapamycin significantly decreased gene expression of LPS-induced *Il6, Il10,* and *Tnfa* and protein concentrations of TNFα and IL6 but maintained higher IL10 levels compared to Veh-DC treated with LPS (Figure 1I–L). Other genes known to be regulated by rapamycin (autophagy or inflammation) are listed in Appendix A. The genes after LPS indicated in green were down-regulated and in red were up-regulated compared to respective vehicle treatment.

### 3.2. Transfer of Rapa-M-DC Protects Kidneys from Ischemic Injury

All DCs were activated with 100 ng/mL LPS prior to transfer in all syngeneic studies (C57BL/6 BMDC→C57BL/6 mice). Half a million DCs were injected 1 day before 26-min bilateral kidney IRI. Schematic of the study design (Figure 2A). As a control, mice were injected with 1 × PBS as no cell (NC) controls. Compared to NC and Veh-DC treated mice, Rapa-M-DC treated mice significantly protected the kidneys from injury, plasma creatinine (Figure 2B); similar changes in BUN were observed in these mice [Sham 41.30 ± 3.4 vs NC 159.75 ± 3.2 vs Veh-DC 151.82 ± 1.5 vs Rapa-M-DC 114.68 ± 16.3; *p* < 0.05 NC or Veh-DC vs Rapa-M-DC]. Morphological changes (Figure 2C,D) paralleled functional studies. Mice treated with Rapa-M-DCs have lower kidney mRNA levels of *Kim1* [Veh-DC 0.016 ± 0.009 vs Rapa-M-DC 0.001 ± 0.002], *Ngal* [Veh-DC 2.14 ± 1.2 vs Rapa-M-DC 0.12 ± 0.09] and *Tnfa* [Veh-DC 0.19 ± 0.15 vs Rapa-M-DC 0.0005 ± 0.0002] compared to Veh-DC treated mice. In these studies, and as demonstrated previously [6,7,17,18], Veh-DC treated mice at a dose of half-million have comparable changes in kidney function and kidney cytokine profile. In syngeneic transfer studies, mice treated with Rapa-M-DCs have reduced apoptosis (terminal deoxynucleotidyl transferase-mediated digoxigenin-deoxyuridine nick-end labeling [TUNEL]) compared to mice treated with Veh-DC (Veh-DC 3.8 ± 1.4 vs Rapa-M-DC 1.1 ± 0.6; *p* < 0.05); representative images from TUNEL analysis are shown in Appendix A. Interestingly, there were no significant changes in number of infiltrating neutrophils in Rapa-M-DC-treated mice compared to Veh-DC (data not shown), suggesting Rapa-M-DC-treated mice may not have any changes in chemokines, although the function of these infiltrating innate immune cells may be altered.

### 3.3. Single Treatment with Rapamycin (Rapa-S) Induces Less Immunogenic DCs without Altering Mitochondrial Dynamics

We next tested if a single treatment with rapamycin was sufficient to induce a regulatory phenotype thus making the feasibility of inducing regulatory DCs phenotype more clinically relevant. WT DCs were isolated and propagated for 8 days in presence of GMCSF, total of three treatments. Seven-day old DCs were treated with rapamycin (10 ng/mL) before treatment with 100 ng/mL LPS for over-night incubation and labeled next day with MitoTracker CMXRos Red (50 nM). Compared to vehicle treatment, Rapa-S treatment does not change mitochondrial content in BMDCs (Figure 3A) with and without LPS. Similarly, there was no change in labeling with MitoTracker Deep Red (100 nM) and MitoSox (5 µM) and after over-night LPS stimulation in Rapa-S-DC compared to Veh-DC (Figure 3B,C). Semi-quantification of histograms MFI for all mitochondrial dyes (Figure 3D). Similarly, to multiple treatment analysis, Veh- and Rapa-S-DCs were staining using done using JC-1 to check if single rapamycin treatment changes the mitochondria membrane potential. Single treatment with rapamycin does not change the baseline mitochondria membrane potential compared to vehicle treated control BMDCs as shown by immunofluorescence (Appendix A) and flow cytometry (Appendix A). LPS reduced oxygen consumption compared to Veh controls in Veh-DCs as expected (Figure 3E, blue to black line). Rapa-S-DCs had no change in basal OCR (Figure 3E, green to blue line at time zero). When ATP production was quantified, LPS reduced ATP production as measured by OCR (blue to black) in Veh-DCs and Rapa-S-DC (Figure 3F). However, Rapa-S-DCs displayed elevated mRNA levels for *Pgc1a* compared to Veh controls, and these levels were maintained in Rapa-S-DCs after LPS (Figure 3G), little to no change was observed in transcript levels of *Tfam*. Next, we tested if single treatment with rapamycin induced a change in DC co-stimulatory molecules after LPS. Interestingly, after LPS treatment, Rapa-S-DC have significantly lower expression levels of co-stimulatory antigen presentation molecules (CD80, CD86, and CD40), MHCII, and PDL1 compared to Veh-DC (Figure 3H). Rapa-S-DCs had a similar expression level for PDL1 compared to Veh-DCs and had similar significant decreases in PDL1/CD86 ratio after LPS stimulation compared to Veh-DCs (Veh/Veh 0.59 ± 0.03 vs Veh/LPS 0.43 ± 0.002, *p* < 0.001 and Rapa-S/Veh 0.69 ± 0.02 vs Rapa-S/LPS 0.56 ± 0.008, *p* < 0.05). Single treatment with Rapamycin significantly reduced gene expression of LPS-induced *Il6, Il10,* and *Tnfa* along with lower protein concentrations of IL10 with little or no change in TNFα and after treated with LPS (Figure 3I–L). Other known genes to be regulated by Rapamycin (autophagy [*Beclin, Atg7, Atg9, Lc3b and Lamp2*] or inflammation [*Il1b, Il12p40, Tlr4, Nos2, Hif1a and Ho1*]) were only partially regulated compared to vehicle DCs after LPS stimulation (Appendix A).

### 3.4. Transfer of Rapa-S-DC Protects Kidneys from Ischemic Injury

All DCs were activated with 100 ng/mL LPS or treated with rapamycin (10 ng/mL) prior to transfer in all syngeneic studies (C57BL/6 BMDC→C57BL/6 mice). Half a million DCs were injected 1 day before bilateral kidney IRI. Schematic of the study design (Figure 4A). As a control, mice were injected with 1 × PBS as no cell (NC) controls. Compared to Veh-DC treated mice, Rapa-S-DC treated mice significantly protected the kidneys from injury plasma creatinine (Figure 4B); similar changes in BUN were observed in these mice [Veh-DC 151.82 ± 1.5 vs Rapa-S-DC 139.53 ± 3.8]. Morphological changes (Figure 4C,D) paralleled functional studies. Mice treated with Rapa-S-DCs have lower kidney mRNA levels of *Kim1* [Veh-DC 0.016 ± 0.009 vs Rapa-S-DC 0.004 ± 0.002], *Ngal* [Veh-DC 2.14 ± 1.2 vs Rapa-S-DC 0.14 ± 0.008] and *Tnfa* [Veh-DC 0.196 ± 0.15 vs Rapa-S-DC 0.0017 ± 0.001] compared to Veh-DC treated mice.

### 3.5. Allogeneic BMDC Transfer of Rapa-M-DC or Rapa-S-DC Equally Protects Kidneys from Ischemic Injury

Half a million DCs were injected 1 day before bilateral kidney IRI for allogeneic (BALB/C BMDC→C57BL/6 mice) transfer studies. Schematic of the study design (Figure 5A). As a control, mice were injected with 1x PBS as no cell (NC) controls. Compared to Veh-DC treated mice, Rapa-M-DC, or Rapa-S-DC treated mice, significantly protected the kidneys from injury (Figure 5B). Morphological changes (Figure 5C,D) paralleled functional studies. Mice treated with rapamycin DCs have lower kidney mRNA levels of *Kim1* [Veh-DC 0.04 ± 0.02 vs Rapa-M-DC 0.005 ± 0.002 vs Rapa-S-DC 0.005 ± 0.0002], *Ngal* [Veh-DC 0.57 ± 0.3 vs Rapa-M-DC 0.19 ± 0.11 vs Rapa-S-DC 0.05 ± 0.01] and *Tnfa* [Veh-DC 0.037 ± 0.02 vs Rapa-M-DC 0.007 ± 0.003 vs Rapa-S-DC 0.008 ± 0.005] compared to Veh-DC treated mice. In allogeneic transfer studies, mice treated with Rapa-M-DC or Rapa-S-DC have reduced apoptosis TUNEL compared to mice treated with Veh-DC (Veh-DC 16.7 ± 8.2 vs Rapa-M-DC 7.2 ± 2.5 vs Rapa-S-DC 9.1 ± 0.1).

### 3.6. Exogenous Mitochondria Loaded DCs (Mito-DC) with Higher Mitochondrial Dynamics and More Immunogenic Phenotype after LPS Stimulation

WT DCs were isolated and propagated for 8 days in presence of GMCSF, total of 3 treatments. Seven-day old DCs were treated over-night with human exogenous mitochondria (10 µg/mL). Compared to vehicle treatment, mitochondria treated DC Mito-DCs have higher mitochondria. Mitochondria were isolated from HEK293 human cell line. The BMDCs were analyzed for the presence of human mitochondria in mouse WT DCs using human mtDNA primers for NADH dehydrogenase (ND). Relative human [mtDNA] content in Veh-DC was 0.26 ± 0.10 versus 16.8 ± 0.59; *p* < 0.001, an increase of 63-fold in genomic DNA isolated from BMDCs. A limitation of the current assay is that it measures human or moue mtDNA relative to nuclear DNA and the small amount of human mtDNA detected in Veh-DC could be due to minor overlap in the sequence of ND in the PCR assay or possibility since we are showing relative expression to mouse nuclear DNA. To determine if higher mitochondrial content also altered mitochondrial function, bioenergetic analysis was undertaken using Seahorse Bioanalyzer. LPS blunted oxygen consumption compared to Veh controls in Veh-DCs as expected (Figure 6A, blue to black line). Mito-DCs have higher basal OCR (Figure 6A, green to blue line at time zero). Upon treatment with uncoupler FCCP, Mito-DC demonstrated a failure to increase maximal respiratory capacity in unstimulated cells that was maintained with LPS stimulation, demonstrating that treatment with isolated mitochondria ablates spare respiratory capacity (likely because already at maximal OCR in basal state). When ATP production was quantified, LPS reduced ATP production as measured by OCR (blue to black) in Veh-DCs. Mito-DC demonstrated significantly greater basal OCR and ATP production compared to Veh-DC in LPS-stimulated DCs (Figure 6B). Mito-DCs displayed elevated mRNA levels for *Pgc1a* and interestingly significantly lower *Tfam* compared to Veh controls (Figure 6C). The differences in *Pgc1a* and *Tfam* in these set of experiments after LPS stimulations compared to data in Figure 1 and Figure 3 are possibly due to use of BALB/c BMDCs. Compared to C57BL/6 BMDCs, DCs from BALB/c have significant changes in mitochondria and cytokine production after stimulation [29,30]. Possibility due to these changes in DCs, historically we have used a higher ischemic time in BALB/c mice compared to C57BL/6 to have similar kidney injury [6,18]. These data indicate that transfer of exogenous healthy mitochondria results in increased mitochondrial content, basal OCR, and ATP production. This suggests the potential for an anti-inflammatory phenotype in DCs after treatment with exogenous mitochondria, as shown in our previous study [18]. Next, we test if DCs treated with exogenous mitochondria that results in increased mitochondrial numbers and function also induced a change in DC co-stimulatory molecules after LPS. Remarkably, with and without LPS stimulation, Mito-DC have significantly higher expression levels of co-stimulatory antigen presentation molecules (CD80, CD86 and CD40), MHCII, and PDL1 compared to Veh-DC (Figure 6D). Furthermore, Mito-DCs had higher expression level for PDL1 compared to Veh-DCs; however, they also had an increase in PDL1/CD86 ratio after LPS stimulation compared to LPS treated Veh-DCs (data not shown). Mito-DC had significantly blunted gene expression of LPS-induced *Tnfa,* but higher gene expression of *Il6* and *Il10* compared to Veh-DC and lower protein concentrations of TNFα and IL10 but maintained higher IL6 levels compared Veh-DC treated with LPS (Figure 6E–H). Other genes known to be regulated by mitochondria (autophagy [*Beclin, Atg7, Atg9, Lc3b and Lamp2*] or inflammation [*Il1b, Il12p40, Tlr4, Nos2, Hif1a and Ho1*]) were partially regulated after LPS stimulation (Appendix A). Veh-DC and Mito-DCs were treated with LPS for 6 and 24 h. Relative changes in mitochondrial complex were measured. Mito-DC treated with and without LPS have significantly higher protein levels of the mitochondrial complexes IV, II, and I and significantly lower levels of Complexes V and III compared to Veh-DCs (Figure 6I). Semi-quantitative analysis of relative protein expression of mitochondrial complexes to GAPDH (Figure 6J). Full length blots are provided as supplementary figures (Appendix A).

### 3.7. Allogeneic BMDC Transfer of Mito-DC Protects Kidneys from Ischemic Injury

Half a million DCs were injected 1 day before bilateral kidney IRI for Mito-DC allogeneic (BALB/C BMDC→C57BL/6 mice) transfer studies. Schematic of the study design (Figure 7A). As control, mice were injected with 1x PBS as no cell (NC) controls. Compared to Veh-DC treated mice, Mito-DC treated mice significantly protected the kidneys from injury (Figure 7B). Morphological changes (Figure 7C,D) paralleled functional studies. Mice treated with Mito-DCs have lower kidney mRNA levels of *Kim1* [Veh-DC 0.04 ± 0.02 vs Mito-DC 0.02 ± 0.01], *Ngal* [Veh-DC 0.57 ± 0.3 vs Mito-DC 0.39 ± 0.20] and *Tnfa* [Veh-DC 0.037 ± 0.02 vs Mito-DC 0.0028 ± 0.0013] compared to Veh-DC treated mice. In allogeneic transfer studies, mice treated with Mito-DC have reduced apoptosis TUNEL compared to mice treated with Veh-DC (Veh-DC 16.7 ± 8.2 vs Mito-DC 3.0 ± 1.2).

## 4. Discussion

In the current study we demonstrated protection from kidney IRI induced by adoptive transfer of rapamycin-treated DCs (either single or multiple times) alternatively regulates DC function that is dependent on relative changes in mitochondria dynamics. Both single and multiple treatment(s) with rapamycin induces reduced immunogenic (lower cytokines and co-stimulatory molecules) DCs. Transfer of either Rapa-M-DCs or Rapa-S-DCs protects the kidneys from ischemic reperfusion injury, indicating the reduced immunogenic state of rapamycin modified DCs induces the protection. Furthermore, transfer of Veh-DC that had transient increases in mitochondria numbers also protects kidneys from IRI (Figure 8A–D)). From these studies we can conclude that injection of less immunogenic DCs is central to inducing protection from IRI a process that can be achieved by pharmacological treatment (rapamycin) or via transfer of healthy exogenous mitochondria. As previously reported [31] and confirmed in our RNAseq studies, overall, there is no clear tolerogenic signature associated with rapamycin-modulated DCs. Although rapamycin DCs do show some significant changes in genes after LPS stimulation compared to vehicle DCs, these were limited to Rapa-M-DCs having lower cytokine-mediated signaling and cytokine production (TNF-signaling via NF-ĸB and IFNγ production) all critical in both innate and adaptive immune responses. As previously reported and confirmed in our study, IL-10 signaling was identified as the major pathway that was different in rapamycin-treated DCs with FDR of 5.87 × 10^−4^ (Appendix A), a hallmark cytokine of tolerogenic signature in DCs.

Historically, tolerogenic DCs (Tol-DC) are identified as being maturation-impaired or semi-immature with low expression of co-stimulatory (CD80, CD86, CD40) with increased IL10 production accompanied by low IL12 and IFNγ secretion, a lower ability to prime T cells and potential to induce Treg after a proinflammatory stimulation [32,33]. Regulatory or tolerogenic DCs can be differentiated in vitro by using immunosuppressant drugs or compounds like FTY720 [18] or rapamycin (the current study), or anti-inflammatory cytokines like IL10 [34] or through genetic modification.

### 4.1. Role of Dendritic Cells in Acute Kidney Injury (AKI)

Although major pharmacological advances have been made for prevention or treatment various injuries none are available for AKI, thus it remains as a major health burden [35]. Furthermore, for preventing or reducing adaptive immune responses mediated allograft rejection or autoimmune disease use non-specific immunosuppressive drugs is associated with adverse side effects. Thus, use of an immune cell type like DCs that can target innate and adaptive immune responses is advantages. Additionally, DCs are important cells in immunity or tolerance, and the idea of using *ex-vivo* regulatory or tolerized DCs in cell-based therapies for cancer, autoimmune disorders and transplantation has been under investigation [36]. Pharmacological (FTY720 [18]) or biological strategies (genetically modified) induce regulatory or tolerogenic DCs (Tol-DC) [37], which are either maturation-resistant or have lower maturation levels or are alternatively activated that ultimately express low levels of MHC I or MHC II and co-stimulatory molecules like CD40, CD80, and CD86. Using CD11c-DTR transgenic mice we have previously published studies that demonstrated that depletion of DCs with diphtheria toxin significantly protected mouse kidneys from ischemia reperfusion injury (IRI) [7,38], and conversely a dose-dependent increase in DC numbers exacerbated kidney injury [7], suggesting that DCs could play a major role in regulating AKI. It is worth mentioning that in AKI, the model of AKI that is utilized (chemotherapy using cisplatin or ischemia) dictates the role DCs play to alter immunological responses. In cisplatin induced AKI, DC ablation either by using transgenic mice or via liposomal chlodronate injection results in more injury as IL-10 produced from DCs in this model is ultimately necessary to dampen nephrotoxic injury [39,40].

### 4.2. Rapamycin: Inflammation, Immune Cells, Ischemia Reperfusion Injury (IRI)

Previously published studies have demonstrated a protective role of rapamycin in AKI models using IRI studies that involved regulating NKT cell infiltration and function thereby ameliorating kidney injury [41], in this study mice were orally treated three times with rapamycin 24 h, 1 h before ischemia and 12 h after ischemia. Chronic oral dosing of mice one day before kidney IRI and every day after (up to 7 days) resulted in higher kidney injury due to inhibition of renal tubular cell proliferation at 1 and 3 days after ischemia [42]. This was evident as in a mouse model of unilateral ureteral obstruction (UUO), where rapamycin treatment ameliorated kidney fibrosis by directly inhibiting mTOR signaling in myofibroblasts and interstitial macrophages [43]. These studies suggest rapamycin may have opposing effects in various acute versus chronic kidney models. However, various studies have reported a detrimental role for rapamycin in animal models of IRI. Oral dosing with rapamycin (4 mg/day) for 7 days in Yorkshire pigs impaired endothelial-dependent vasorelaxation and increased myocardial necrosis in a model of coronary artery occlusion [44]. However, in mice, acute treatment with rapamycin (one hour before ischemia, intraperitoneal injection) induced cardio-protection by regulating JAK2-STAT3 signaling to protect against myocardial infarction [45]. Depending on the route (gavage or subcutaneous or intraperitoneal), dosing (0.1–6 mg/kg/day), timing (early or delayed) [46], and type of model used (heart [47], liver [48] or kidney [41,49]), rapamycin has been shown to be protective [47] or not protective [50].

### 4.3. Cell Therapy and Rapamycin

The propagation of mouse BMDCs or human CD14^+^ monocytes that we consider chronic induced an alternative protective or regulatory phenotype in DCs that regulated both innate and adaptive immune responses. In lethal graft-versus-host disease, DCs generated with rapamycin had decreased MHC II and co-stimulatory molecules but had a higher population of IL12 producing after LPS stimulation. These IL12 producing Rapa-M-DC enhanced alloreactive T cell apoptosis, a mechanism that involved IFNγ [51]. Similarly, CD14^+^ monocytes propagated in the presence of rapamycin also produced higher IL12p70 along with IL27 levels that regulated NK cell function, thus changing allogeneic T cell responses through IFNγ [52]. In kidney, transfer of rapamycin-treated Tregs had markedly suppressed T cell, myeloid cell, and myofibroblast responses, resulting in improved acute and chronic kidney disease compared to direct treatment of mice with Rapamycin [53]. Similarly, adoptive transfer of rapamycin-treated myeloid-derived suppressor cells (MDSCs) into mice improved kidney function with diminished histological damages, and immune cell infiltration [54]. These studies provide evidence that cell therapy that involves treatment of cells with rapamycin prior to injection avoids adverse side effects that might be associated with systemic rapamycin treatment. Additionally, we have tested the use of metformin or compounds like 5-aminoimidazole-4-carboxamide-1-β-4-ribofuranoside (AICAR) to induce a regulatory phenotype in mouse DCs to test their role in altering kidney IRI. Like the dosing strategy used with rapamycin (Rapa-S-DC), we treated BMDCs with low doses of metformin (1 mM) or AICAR (1 mM) at day 7 along with LPS. We systemically injected 0.5 x 10^6^ of either metformin-DCs or AICAR-DCs one day before kidney IRI. Mice treated with either metformin-DCs or AICAR-DCs were significantly protected compared to either NC or Veh-DCs treated mice (unpublished observations, Bajwa lab). Thus, multiple drugs or compounds that induce metabolic changes or activate the AMPK pathway might be worth testing by either alone or in combination. It is worth noting that using multiple combination of drugs or compounds (metformin or AICAR) that could potentially complement the protecting effects of rapamycin is yet to be explored when evaluating their use in BMDCs. In cancer cells, combining AICAR with rapamycin enhances the efficacy of rapamycin to further suppress mTORC2 to induce apoptosis [55]. In T cells, combination use of metformin and rapamycin has also been tested as potential metabolic checkpoint regulator to influence T cell function [56]. Thus, these combination therapeutic modalities should be tested in the future with tolerogenic or regulatory BMDCs as potential direct or indirect regulators of the metabolic checkpoints.

### 4.4. Rapamycin: Role of Mitochondria in Dendritic Cells and Macrophages

DC and macrophage functions are regulated by mitochondrial metabolism. Type 1 macrophages [57,58] and immunogenic DCs (TLR-mediated activation) have metabolic transition in which mitochondrial oxidative phosphorylation is inhibited by endogenously synthesized nitric oxide (NO) and commitment to high aerobic glycolytic rates [59]. The activation of DCs or macrophages by several TLR agonists (LPS or CpG) leads to a rapid increase in glycolysis, followed by a decrease in OXPHOS and mitochondrial membrane potential [59,60,61]. Some role for mitochondria has been demonstrated with DCs treated *in-vitro* with vitamin D; these DCs have increased OXPHOS, mitochondrial mass, and mROS production [62], our recent study using FTY720 [18] showed similar changes in DC (higher OXPHOS and less immunogenicity), acute treatment with rapamycin has been shown to extend the cellular lifespan of DCs that was dependent of preservation of mitochondrial function [10] through inhibition of NO. Contrary to increasing the life span of DCs, the same groups also reported that acute or brief exposure to rapamycin in DCs also makes them potent activators of antigen specific CD8+ T cells, thus enhancing control of B16 melanoma as vaccination in mice [63]. Acute treatment (90 min) of human monocyte derived DCs with rapamycin suppresses LPS induced immunostimulatory molecules and allo-stimulatory potential that involved the inability of treated DCS to augment NF-ĸB signaling [64]. In macrophages, rapamycin suppresses the production of IL1β and IL-18 after LPS stimulation, a process that also involves reducing mitochondrial ROS (mtROS) by directly inhibiting NLRP3 inflammasome-p38 MAPK-NF-ĸB pathways [65]. rapamycin-conditioned DC (Rapa-DC) from mice and humans behave differently, compared to mouse Rapa-DC, human Rapa-DC have proven to be partially resistant to maturation after proinflammatory cytokines and display heterogeneity in regulating effector T-cell expansion and function [66]. Much of the immunoregulatory properties of rapamycin-conditioned monocyte-derived DCs and their role in transplantation has been elegantly reviewed by Macedo et al. [67].

Our current findings indicate that single or multiple rapamycin treated DCs have a reduced immunogenic and immunostimulatory phenotype after LPS stimulation. Transfer of these DCs protects kidneys from IRI. Rapamycin alternatively regulates mitochondria dynamics; only multiple treatment induces higher mitochondria numbers. These data suggest that DC mitochondrial dynamics (higher mitochondria content), along with less immunogenic phenotype induced by rapamycin, cooperatively protect kidneys from ischemic injury. Although more interestingly, our data with exogenous mitochondria treated DCs indicates that higher mitochondria numbers can overcome the more immunogenic state of DCs after LPS stimulation to also protect kidneys from IRI. It will be interesting to confirm if higher mitochondria numbers in DCs alter T cell responses, DCs do have the ability to alter immune response as our previous study indicated. Our preliminary data indicates, in allogenic T cell proliferation studies, DCs with higher mitochondria numbers do suppress CD4 T cell proliferation in mixed lymphocyte reactions (MLR, data not shown). It is currently unclear if DCs with higher mitochondrial numbers can induce a Treg response that could positively contribute to protecting kidneys from IRI. These will be the focus of our next studies.

## 5. Conclusions

In summary, we have demonstrated that BMDCs can regulate kidney injury by a mechanism that involves mitochondria biogenesis either by multiple treatments with rapamycin or exogenously increasing mitochondria numbers by using exogenous mitochondria. We conclude that regulatory DCs (DCs with higher mitochondria numbers) may be useful in kidney IRI as well as in other inflammatory states such as transplantation and autoimmune disorders to prevent or treat injury.

## Figures and Tables

**Figure 1 ijms-22-05386-f001:**
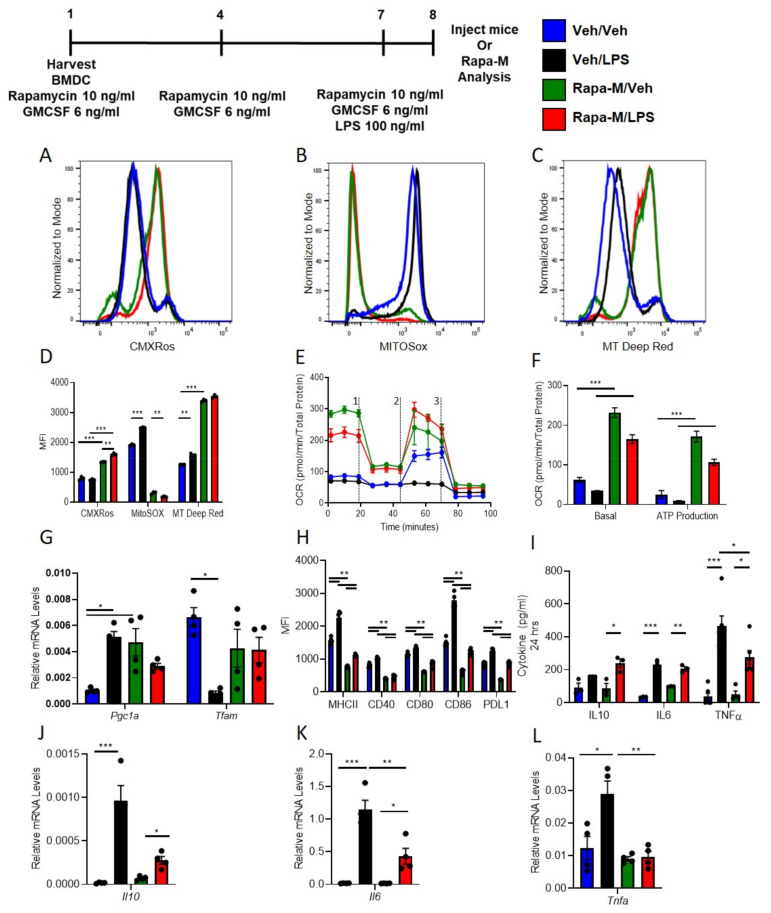
Rapamycin treatment induces less immunogenic DCs that maintain higher mitochondrial dynamics after LPS stimulation. Histograms of 8-day old Veh-DC and Rapa-M-DC were treated with 100 ng/mL LPS for 24 h or left unstimulated (Veh) and labeled with (**A**) MitoTracker CMXRos (50 nM) or (**B**) MitoSox (5 µM) or (**C**) MitoTracker Deep Red (100 nM). (**D**) Semi-quantified analysis of MFI of mitochondria dyes labeled flow cytometry data. (**E**) DCs were seeded in a Seahorse XF-24e analyzer, stimulated with and without LPS for 24 h, and oxygen consumption rate (OCR) was determined during sequential treatments with oligomycin (1), FCCP (2) and antimycin A/rotenone (3). (**F**) Quantification of basal OCR and ATP production. (**G**) mRNA levels of *Pgc1a* and *Tfam* in Veh-DC and Rapa-M-DC treated with 100 ng/mL LPS or left unstimulated for 24 h. (**H**) Flow cytometry analysis of MHC II, co-stimulatory molecules (CD40/80/86) and PDL1 expression was determined with and without LPS stimulation. (**I**) ELISA of IL10, IL6, and TNFα from the Veh-DC and Rapa-M-DC treated with 100 ng/mL LPS for 24 h. (**J**–**L**) mRNA levels of *Il10, Tnfa, and Il6* in Veh-DC and Rapa-M-DC treated with 100 ng/mL LPS or left unstimulated for 24 h. * *p* ≤ 0.05, ** *p* ≤ 0.01, and *** *p* ≤ 0.001, one-way ANOVA followed by Tukey’s post-test. Data represent means ± SEM of triplicates.

**Figure 2 ijms-22-05386-f002:**
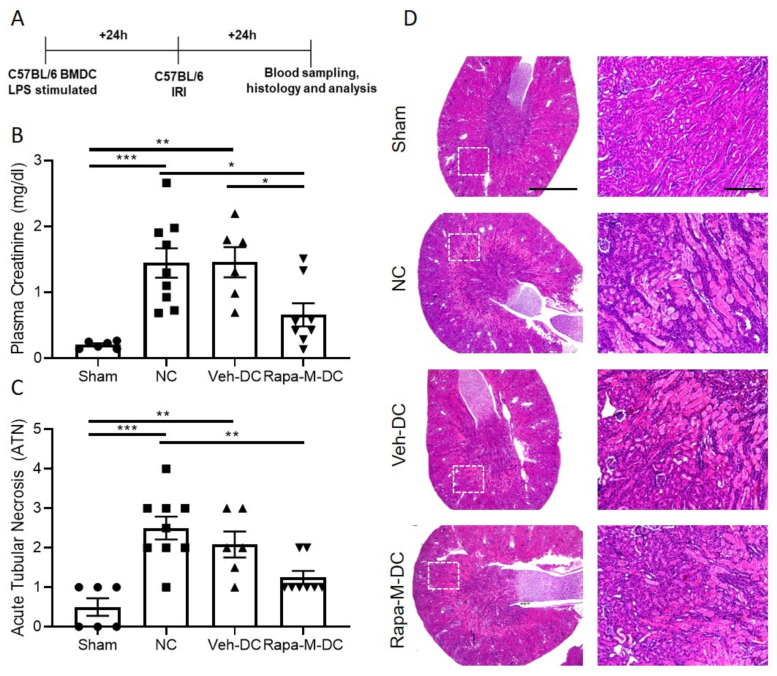
Pretreatment with Rapa-M-DCs protects kidneys from ischemia reperfusion injury in syngeneic transfer studies. Mice were intravenously (i.v.) injected with 0.5 × 10^6^ DCs (Veh-DC or Rapa-M-DC) that were LPS stimulated over-night and as control no cells (NC) one day before bilateral kidney IRI. (**A**) Protocol for experimental setup. (**B**) Plasma Creatinine (**C**,**D**) Quantification of acute tubular injury (ATN) and renal histology (H&E). Scale bar (whole kidney section), 1000 µm and box inset, 200 µm. Data represent means ± SEM, * *p* ≤ 0.05, ** *p* ≤ 0.01, and *** *p* ≤ 0.001 one-way ANOVA followed by Tukey’s post-test.

**Figure 3 ijms-22-05386-f003:**
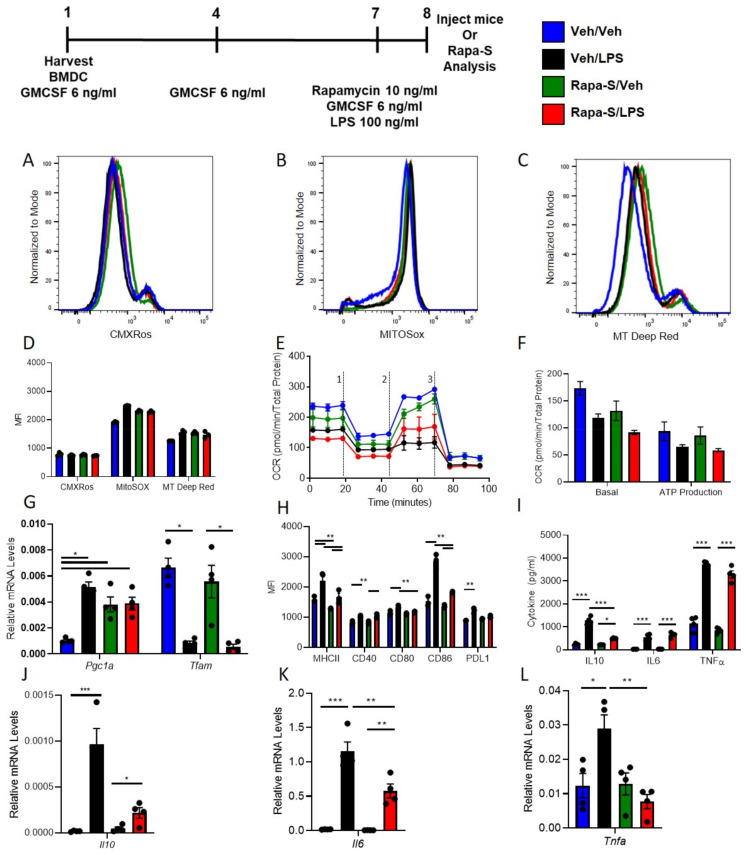
Single rapamycin, Rapa-S treatment induces less immunogenic DCs without higher mitochondrial dynamics after LPS stimulation. Histograms of 8-day old Veh-DC and Rapa-S-DC were treated with 100 ng/mL LPS for 24 h or left unstimulated (Veh) and labeled with (**A**) MitoTracker CMXRos (50 nM) or (**B**) MitoSox (5 µM) or (**C**) MitoTracker Deep Red (100 nM). (**D**) Semi-quantified analysis of MFI of mitochondria dyes labeled flow cytometry data. (**E**) DCs were seeded in a Seahorse XF-24e analyzer, stimulated with and without LPS for 24 h, and oxygen consumption rate (OCR) was determined during sequential treatments with oligomycin (1), FCCP (2) and antimycin A/rotenone (3). (**F**) Quantification of basal OCR and ATP production. (**G**) mRNA levels of Pgc1a and Tfam in Veh-DC and Rapa-S-DC treated with 100 ng/mL LPS or left unstimulated for 24 h. (**H**) Flow cytometry analysis of MHC II, co-stimulatory molecules (CD40/80/86), and PDL1 expression were determined with and without LPS stimulation. (**I**) ELISA of IL10, IL6, and TNFα from the Veh-DC and Rapa-S-DC treated with 100 ng/mL LPS for 24 h. (**J**–**L**) mRNA levels *of Il10, Tnfa*, and *Il6* in Veh-DC and Rapa-S-DC treated with 100 ng/mL LPS or left unstimulated for 24 h. * *p* ≤ 0.05, ** ≤ 0.01, and *** *p* ≤ 0.001, one-way ANOVA followed by Tukey’s post-test. Data represent means ± SEM of triplicates.

**Figure 4 ijms-22-05386-f004:**
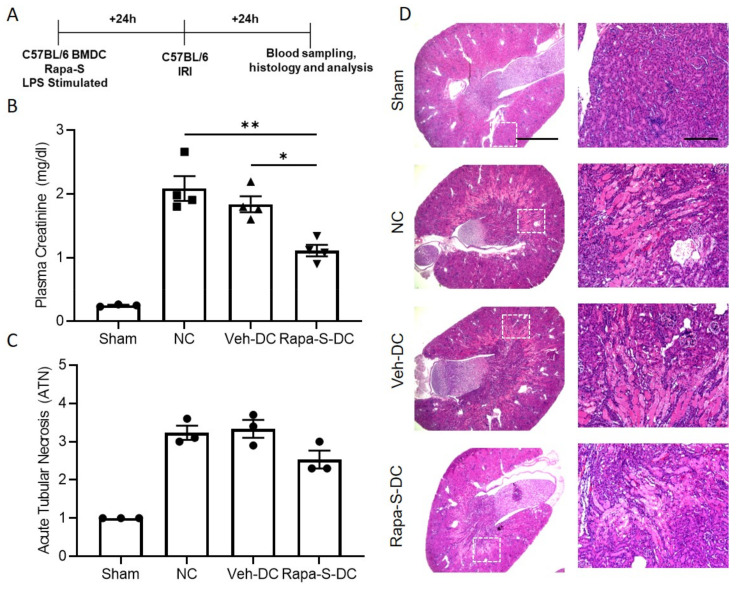
Pretreatment with Rapa-S-DC protects kidneys from ischemia reperfusion injury in syngeneic transfer studies. Mice were intravenously (i.v.) injected with 0.5 × 10^6^ DCs (Veh-DC or Rapa-S-DC) that were LPS stimulated over-night and as control no cells (NC) one day before bilateral kidney IRI. (**A**) Protocol for experimental setup. (**B**) Plasma creatinine (**C**,**D**) Quantification of acute tubular injury (ATN) and renal histology (H&E). Scale bar (whole kidney section), 1000 µm and box inset, 200 µm. Data represent means ± SEM, * *p* ≤ 0.05, and ** *p* ≤ 0.01, one-way ANOVA followed by Tukey’s post-test.

**Figure 5 ijms-22-05386-f005:**
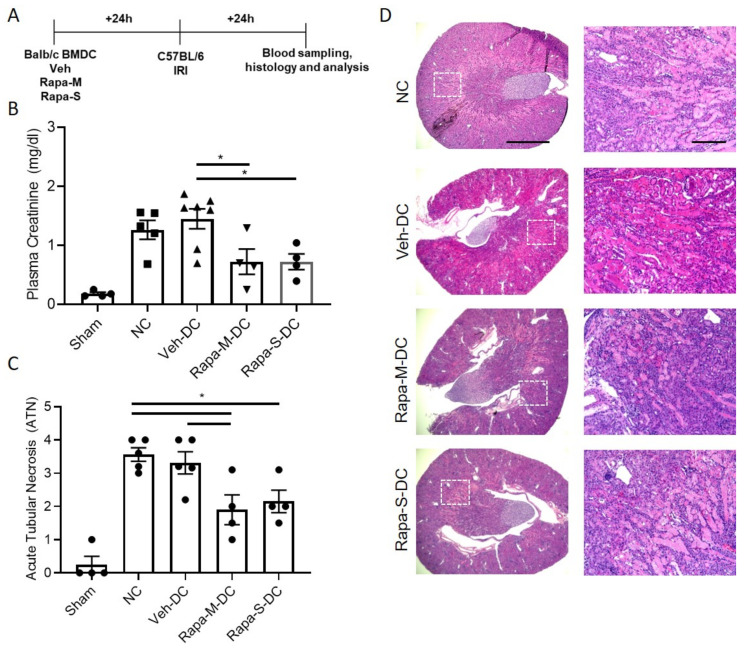
Pretreatment with either Rapa-M-DC or Rapa-S-DC protects kidneys from ischemia reperfusion injury in allogeneic transfer studies. Mice were intravenously (i.v.) injected with 0.5 × 10^6^ DCs (Veh-DC or Rapa-M-DC or Rapa-S-DC) and as control no cells (NC) one day before bilateral kidney IRI. (**A**) Protocol for experimental setup. (**B**) Plasma Creatinine (**C**,**D**) Quantification of acute tubular injury (ATN) and renal histology (H&E). Scale bar (whole kidney section), 1000 µm and box inset, 200 µm. Data represent means ± SEM, * *p* ≤ 0.05, one-way ANOVA followed by Tukey’s post-test.

**Figure 6 ijms-22-05386-f006:**
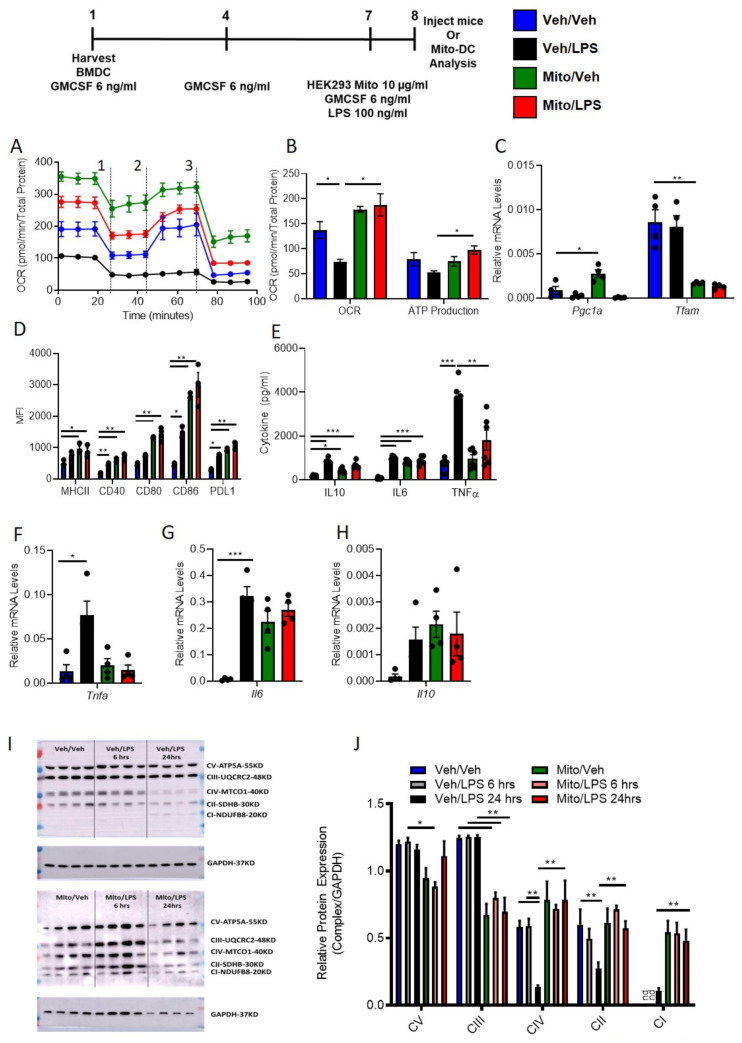
Mitochondria loaded DCs (Mito-DC) have higher mitochondrial function and immunogenicity after LPS stimulation compared to Veh-DCs. Seven-day old Veh-DC and were incubated with 10 µg/mL HEK293 mitochondria one day before treatment with 100 ng/mL LPS for additional 24 h or left unstimulated (Veh). (**A**) 24 h after addition of mitochondria, DCs were seeded in a Seahorse XF-24e analyzer, stimulated with and without LPS for 24 h, and oxygen consumption rate (OCR) was determined during sequential treatments with oligomycin (1), FCCP (2) and antimycin A/rotenone (3). (**B**) Quantification of basal OCR and ATP production. (**C**) mRNA levels of *Pgc1a* and *Tfam* in Veh-DC and Mito-DC treated with 100 ng/mL LPS or left unstimulated for 24 h. (**D**) Flow cytometry analysis of MHCII, co-stimulatory molecules (CD40/80/86) and PDL1 expression was determined with and without LPS stimulation. (**E**) ELISA of IL10, IL6, and TNFα from the Veh-DC and Mito-DC treated with 100 ng/mL LPS for 24 h. (**F**–**H**) mRNA levels of *Tnfa, Il6*, and *Il10* in Veh-DC and Mito-DC treated with 100 ng/mL LPS or left unstimulated for 24 h. Data represent means ± SEM, * *p* ≤ 0.05, ** *p* ≤ 0.01, and *** *p* ≤ 0.001, one-way ANOVA followed by Tukey’s post-test. Data represent means ± SEM of triplicates.

**Figure 7 ijms-22-05386-f007:**
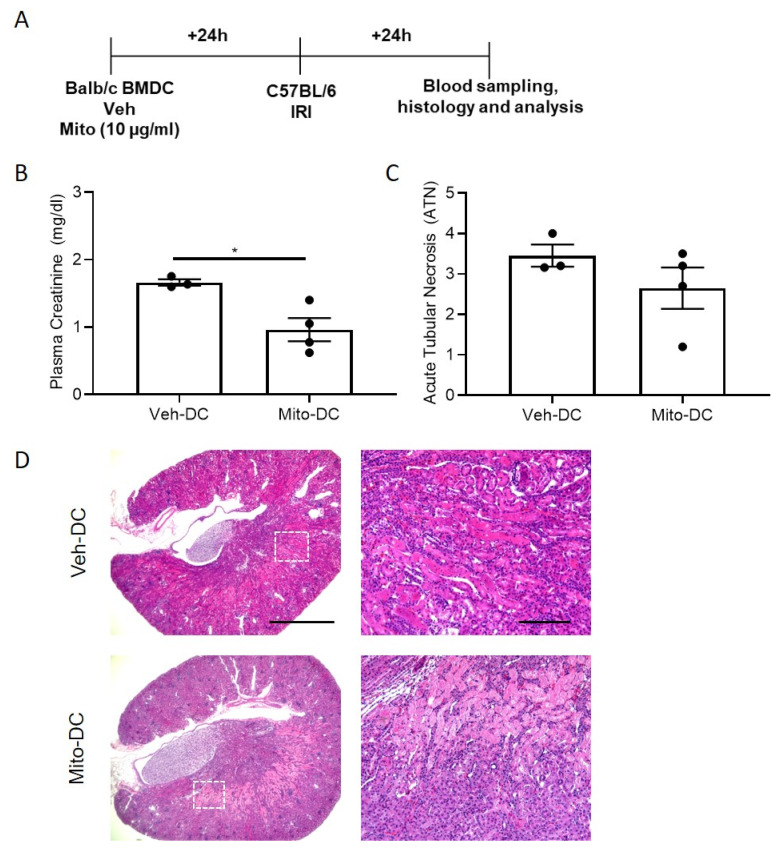
Pretreatment with either Mito-DC protects kidneys from ischemia reperfusion injury in allogeneic transfer studies. Mice were intravenously (i.v.) injected with 0.5 × 10^6^ DCs (Veh-DC or Mito-DC) one day before bilateral kidney IRI. (**A**) Protocol for experimental setup. (**B**) Plasma creatinine (**C**,**D**) Quantification of acute tubular injury (ATN) and renal histology (H&E). Scale bar (whole kidney section), 1000 µm and box inset, 200 µm. * *p* ≤ 0.05, student t test. Data represent means ± SEM, Unpaired t-test.

**Figure 8 ijms-22-05386-f008:**
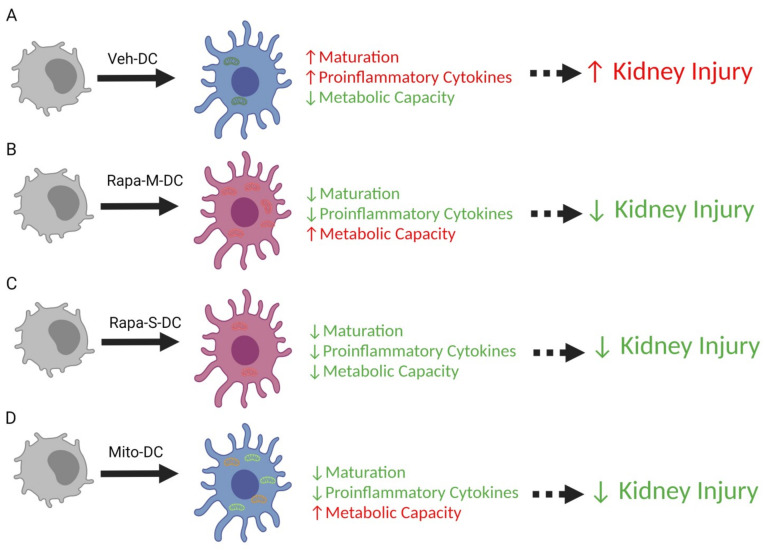
Model summarizing the change in bone marrow dendritic cells conditioned with rapamycin or treated with exogenous mitochondria protect kidneys from ischemic reperfusion injury. (**A**) Dendritic cells propagated for 8 days in GMCSF have higher maturation (MHCII, CD40, CD80, CD86) and pro-inflammatory cytokines (TNFα, IL1β, IL12p40, IL6) and lower metabolic capacity (mitochondrial oxygen consumption rate) after overnight treatment with 100 ng/mL LPS stimulation. Systemic transfer of 0.5 × 10^6^ DCs one day before bilateral kidney IRI results in higher kidney injury. (**B**) Dendritic cells propagated for 8 days in GMCSF and three treatments with rapamycin have lower maturation (MHCII, CD40, CD80, CD86) and proinflammatory cytokines (TNFα, IL1β, IL12p40, IL6) and higher metabolic capacity (mitochondrial oxygen consumption rate) after overnight treatment with 100 ng/mL LPS stimulation. Systemic transfer of 0.5 × 10^6^ DCs one day before bilateral kidney IRI results in lower kidney injury compared to mice treated with PBS or Veh-DC (**A**). (**C**) Dendritic cells propagated for 8 days in GMCSF and 1 overnight treatment with rapamycin have lower maturation (MHCII, CD40, CD80, CD86) and proinflammatory cytokines (TNFα, IL1β, IL12p40, IL6) and lower metabolic capacity (mitochondrial oxygen consumption rate) after overnight treatment with 100 ng/mL LPS stimulation. Systemic transfer of 0.5 × 10^6^ DCs one day before bilateral kidney IRI results in lower kidney injury compared to mice treated with PBS or Veh-DC (**A**). (**D**) Dendritic cells propagated for 8 days in GMCSF and one overnight treatment with exogenous mitochondria have lower maturation (MHCII, CD40, CD80, CD86) and proinflammatory cytokines (TNFα, IL1β, IL12p40, IL6) and higher metabolic capacity (mitochondrial oxygen consumption rate) after overnight treatment with 100 ng/mL LPS stimulation. Systemic transfer of 0.5 × 10^6^ DCs one day before bilateral kidney IRI results in lower kidney injury compared to mice treated with PBS or Veh-DC (**A**). The figure was created by the authors with BioRender.com.

## Data Availability

The data discussed in this publication have been deposited in NCBI’s Gene Expression Omnibus and are accessible through GEO series accession number GSE158161 (https://www.ncbi.nlm.nih.gov/geo/query/acc.cgi?acc=GSE158161, accessed on 20 April 2021).

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
