# Peer review of "Rapamycin Alternatively Modifies Mitochondrial Dynamics in Dendritic Cells to Reduce Kidney Ischemic Reperfusion Injury"

_ijms, 2021, doi:10.3390/ijms22105386_

Round 1
Reviewer 1 Report
Bajwa's group elaborated the role of rapamycin in mitochondrial dynamics which has tremendous impact to improve therapeutic index for kidney injury. Few things need to be addressed before it is ready for acceptance. They are as follows:
- It has been discussed (doi: 10.1158/1535-7163.MCT-15-0720) how different doses of rapamycin plays variant roles in signaling pathways. This should be discussed in the introduction part.
- It has also been shown that AMPK activator metformin also plays a role in AKI (DOI: 10.34172/npj.2021.13) while it has also been shown that AMPK activator AICAR plays a synergistic role with rapamycin in case of improving therapy purposes (DOI: 10.1080/15384101.2015.1087623). Since authors have discussed the role of rapamycin in this manuscript, it will be intriguing to know their perspective on utilizing any AMPK activator along with rapamycin to improve the therapy treatment. This can be discussed in discussion part as one of the future direction. It will be an important aspect of this current research.
- Also mTOR plays a definite role in metabolic checkpoints(doi: 10.18632/oncoscience.253). Authors can add few lines on their point of view how metabolism/ metabolic checkpoints might play a role in this current study. It will be important viewpoint which should be discussed in the discussion part.
- Lastly, authors should add a model depicting their take home message from this manuscript. This will sum up the whole concept and finding discussed in this current manuscript.
Author Response
Thank you for the opportunity to revise our manuscript.
Reviewer 1
Comments and Suggestions for Authors: Bajwa's group elaborated the role of rapamycin in mitochondrial dynamics which has tremendous impact to improve therapeutic index for kidney injury. Few things need to be addressed before it is ready for acceptance. They are as follows:
- It has been discussed (doi: 10.1158/1535-7163.MCT-15-0720) how different doses of rapamycin plays variant roles in signaling pathways. This should be discussed in the introduction part.
*** Thank you for pointing out this paper. We have included a sentence about dosing in our introduction (changes shown in red) and updated our references accordingly. The changes can be found on page 5.
- It has also been shown that AMPK activator metformin also plays a role in AKI (DOI: 10.34172/npj.2021.13) while it has also been shown that AMPK activator AICAR plays a synergistic role with rapamycin in case of improving therapy purposes (DOI: 10.1080/15384101.2015.1087623). Since authors have discussed the role of rapamycin in this manuscript, it will be intriguing to know their perspective on utilizing any AMPK activator along with rapamycin to improve the therapy treatment. This can be discussed in discussion part as one of the future direction. It will be an important aspect of this current research.
**We have updated the discussion to include information about metformin and AICAR and potential use of combination therapy. The discussion was also updated to include some of our unpublished observations using metformin and AICAR to induce a regulatory phenotype in DC for testing in kidney ischemia reperfusion injury. The updates can be found in red on page 24.
- Also mTOR plays a definite role in metabolic checkpoints(doi: 10.18632/oncoscience.253). Authors can add few lines on their point of view how metabolism/ metabolic checkpoints might play a role in this current study. It will be important viewpoint which should be discussed in the discussion part.
** We agree with the reviewer that there is a role for mTOR as metabolic checkpoint and we update the discussion. There is however very limited information about how metabolic checkpoints inhibitors can induce a change in phenotype in BMDCs. The updated discussion to include this information can be found in red on page 24.
- Lastly, authors should add a model depicting their take home message from this manuscript. This will sum up the whole concept and finding discussed in this current manuscript.
** We have included a new figure 8 to show a model depicting the overall take home message.
Reviewer 2 Report
Reviewing the manuscript entitled, “Rapamycin alternatively modifies mitochondrial dynamics in dendritic cells to reduce kidney ischemic reperfusion injury” by Namwanje M er al., this is an article focusing on relevance between rapamycin treated DCs and modification of mitochondrial dynamics. Although this paper has undergone numerous experiments and is a very interesting study, the progress of the manuscript is jerky, and it is hard to understand what is new evidence as a result of this study. The authors need to response the following concerns to reach to acceptable quality.
Concerns
Your results showed the clearly difference between one time and three times treatments of rapamycin. If so, you should state why the number of multiple treatments is 3 or take a titration. Is three times treatment adequate?
In the animal experiments, why did not you experiment with rapamycin treatment without LPS stimulation? In general, LPS stimulation is performed to produce mature DC, and high expression of MHC II and CD86 confirms maturity. However, in the discussion at page 13, you mentioned “Pharmacological (FTY720) or biological strategies (genetically modified) induce regulatory or tolerogenic DCs (Tol-DC), which are either maturation-resistant or have lower maturation levels, or are alternatively activated that ultimately express low levels of MHC I or MHC II and co-stimulatory molecules”. From these points of view and the results shown in Figure 1, DCs treated with rapamycin and without LPS stimulation appear to be the best.
The authors need to describe a causal relationship among AKI, DC, rapamycin treatment, LPS stimulation with figure for easily understandable.
The authors need to add an expression level of IL 12.
The animal experiments in Figure 4 are incomplete. The authors need to add results of sham operation.
The authors need to modify English including font size.
Author Response
Thank you for the opportunity to revise our manuscript. Your time and input is appreciated.
Reviewer 2
Comments and Suggestions for Authors
Reviewing the manuscript entitled, “Rapamycin alternatively modifies mitochondrial dynamics in dendritic cells to reduce kidney ischemic reperfusion injury” by Namwanje M er al., this is an article focusing on relevance between rapamycin treated DCs and modification of mitochondrial dynamics. Although this paper has undergone numerous experiments and is a very interesting study, the progress of the manuscript is jerky, and it is hard to understand what is new evidence as a result of this study.
The authors need to response the following concerns to reach to acceptable quality.
Concerns
- Your results showed the clearly difference between one time and three times treatments of rapamycin. If so, you should state why the number of multiple treatments is 3 or take a titration. Is three times treatment adequate?
** The rational for choosing the three-rapamycin treatment was due to ours and others previously published work. Prior to staring the studies, we tested different doses of rapamycin and settled on 10 ng/ml as this induced a significant change in costimulatory molecules after LPS treatment compared to vehicle treated DCs after LPS stimulation. The dosing was limited to 3 as we have, and others have historically used 6-8 day old propagated BMDCs. The time frame is also based on the expression and purity of the culture as tested by the marker CD11c. Additionally, the 3 times treatment time points were chosen to coincide with GMCSF treatments. To answer the reviewers second question about “is three times treatment adequate”, our data demonstrates that a single treatment with rapamycin is adequate to induce a regulatory phenotype in BMDCs (Figures 3-5). The three times was needed to induce a change in mitochondria.
- In the animal experiments, why did not you experiment with rapamycin treatment without LPS stimulation? In general, LPS stimulation is performed to produce mature DC, and high expression of MHC II and CD86 confirms maturity. However, in the discussion at page 13, you mentioned “Pharmacological (FTY720) or biological strategies (genetically modified) induce regulatory or tolerogenic DCs (Tol-DC), which are either maturation-resistant or have lower maturation levels or are alternatively activated that ultimately express low levels of MHC I or MHC II and co-stimulatory molecules”. From these points of view and the results shown in Figure 1, DCs treated with rapamycin and without LPS stimulation appear to be the best.
- **The rational to treat the DCs for syngeneic (C57BL/6 DCs->C57BL/6 mice) transfer studies is due to our previously published work that demonstrates that at a dose of half-million if we do a subthreshold ischemia we do not get an added response with activated DCs. Also, at a standard clamp time of ischemia we can have a dose dependent increase in injury with Veh-DCs (PMID: 22855711). We agree with the reviewer comment and have therefore also used DCs that were not stimulated with LPS (BALB/c DCs->C57BL/6 mice), this data can be found in figure 5.
- The authors need to describe a causal relationship among AKI, DC, rapamycin treatment, LPS stimulation with figure for easily understandable.
** We have included new figure 8 to describe the overall finding of our current work to better demonstrate the causal relationship between AKI, DCs and Rapamycin.
- The authors need to add an expression level of IL 12.
** We have included the gene expression for IL12p40 in our supplemental tables 2 and 3 and the manuscript under results have been updated to include the list of genes shown in supplemental tables, these changes are marked in red on pages ## and ##.
- The animal experiments in Figure 4 are incomplete. The authors need to add results of sham operation.
** We apologize for this mistake. Figure 4 has been updated to include the missing values for sham mice.
- The authors need to modify English including font size.
** We have increased the font size to 12.
Round 2
Reviewer 1 Report
All concerns have been addressed. Ready for acceptance.
Reviewer 2 Report
No further comments. the revising manuscript is acceptable quality.